# IMAGE TRANSFORMER

## ABSTRACT

Image generation has been successfully cast as an autoregressive sequence generation or transformation problem. Recent work has shown that self-attention is an effective way of modeling textual sequences. In this work, we generalize a recently proposed model architecture based on self-attention, the Transformer, to a sequence modeling formulation of image generation with a tractable likelihood. By restricting the self-attention mechanism to attend to local neighborhoods we significantly increase the size of images the model can process in practice, despite maintaining significantly larger receptive fields per layer than typical convolutional neural networks. We propose another extension of self-attention allowing it to efficiently take advantage of the two-dimensional nature of images.

While conceptually simple, our generative models trained on two image data sets are competitive with or significantly outperform the current state of the art in autoregressive image generation on two different data sets, CIFAR-10 and ImageNet.

We also present results on image super-resolution with a large magnification ratio, applying an encoder-decoder configuration of our architecture. In a human evaluation study, we show that our super-resolution models improve significantly over previously published autoregressive super-resolution models. Images they generate fool human observers three times more often than the previous state of the art.

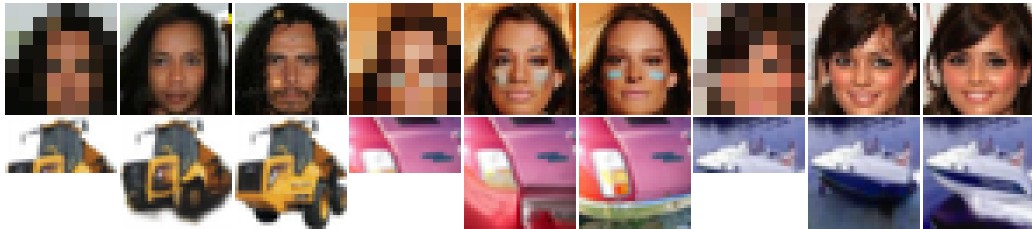

Table 1: Three outputs of a CelebA super-resolution model followed by three image completions by a conditional CIFAR-10 model, with input, model output and the original from left to right

## 1 INTRODUCTION

Recent advances in modeling the distribution of natural images with neural networks allow them to generate increasingly natural-looking images.

Some models, such as the PixelRNN and PixelCNN (van den Oord et al., 2016), have a tractable likelihood. Beyond licensing the comparatively simple and stable training regime of directly maximizing log-likelihood, this enables the straightforward application of these models in problems such as image compression (van den Oord & Schrauwen, 2014) and probabilistic planning and exploration (Bellemare et al., 2016).

The likelihood is made tractable by modeling the joint distribution of the pixels in the image as the product of conditional distributions (Larochelle & Murray, 2011; Theis & Bethge, 2015). Having

---

Code available at `anonymized`

thus turned the problem into a sequence modeling problem, the state of the art approaches apply recurrent or convolutional neural networks, predicting each next pixel given all previously generated pixels (van den Oord et al., 2016). Training recurrent neural networks to sequentially predict each pixel of even a small image is computationally very challenging. Thus, models based on much more parallelizable convolutional neural networks such as the PixelCNN have recently received much more attention, and have now surpassed the PixelRNN in quality PixelCNN.

One disadvantage of CNNs compared to RNNs is their typically fairly limited receptive field. This can adversely affect their ability to model long-range phenomena common in images, such as symmetry and occlusion, especially with a small number of layers. Growing the receptive field has been shown to improve quality significantly (Salimans et al.). Doing so, however, like deepening the network, comes at a significant cost in number of parameters and consequently computational performance and can make training such models more challenging.

In this work we aim to find a better balance in the trade-off between the virtually unlimited receptive field of the necessarily sequential PixelRNN and the limited receptive field of the much more parallelizable PixelCNN and its various extensions.

We adopt similar factorizations of the joint pixel distribution as previous work. Following recent work on modeling text (Vaswani et al., 2017), however, we propose eschewing recurrent and convolutional networks in favor of the Image Transformer, a model based entirely on a self-attention mechanism (Cheng et al., 2016; Parikh et al., 2016). The specific, locally restricted form of multi-head self-attention we propose could also be interpreted as a sparsely parameterized form of gated convolution, allowing for significantly larger receptive fields than CNNs at the same number of parameters.

Despite comparatively low resource requirements for training, the Image Transformer attains a new state of the art in modeling images from the standard ImageNet data set, as measured by log-likelihood. Our experiments indicate that increasing the size of the receptive field plays a significant role in this improvement.

Many applications of image density models require conditioning on additional information of various kinds: from images in enhancement or reconstruction tasks such as super-resolution, in-painting and denoising to text when synthesizing images from natural language descriptions (Mansimov et al., 2015). In visual planning tasks, conditional image generation models could predict future frames of video conditioned on previous frames and taken actions.

In this work we hence also evaluate two different methods of performing conditional image generation with the Image Transformer. In image-class conditional generation we condition on an embedding of one of a small number of image classes. In super-resolution with high magnification ratio, we condition on a very low-resolution image, employing the Image Transformer in an encoder-decoder configuration (Kalchbrenner & Blunsom, 2013). In comparison to recent work on autoregressive super-resolution (Dahl et al., 2017), a human evaluation study found images generated by our models look convincingly natural significantly more often.

## 2 BACKGROUND

There is a broad variety of types of image generation models in the literature. This work is most strongly inspired by autoregressive models such as fully visible belief networks and NADE (Bengio & Bengio, 2000; Larochelle & Murray, 2011) in that we also factor the joint probability of the image pixels into conditional distributions. Following PixelRNN (van den Oord et al., 2016), we also model the color channels of the output pixels as discrete values generated from a multinomial distribution, implemented using a simple softmax layer.

The current state of the art in modeling images in the CIFAR-10 data set was achieved by the PixelCNN++, modeling the output pixel distribution with a discretized logistic mixture likelihood, conditioning on whole pixels instead of color channels and changes to the architecture (Salimans et al.). Most of these modifications can also be applied to our model which we plan to evaluate in future work.

Another, currently wildly popular direction of research in image generation is training models with an adversarial loss (Goodfellow et al., 2014). Typically, in this regime a generator network is trained

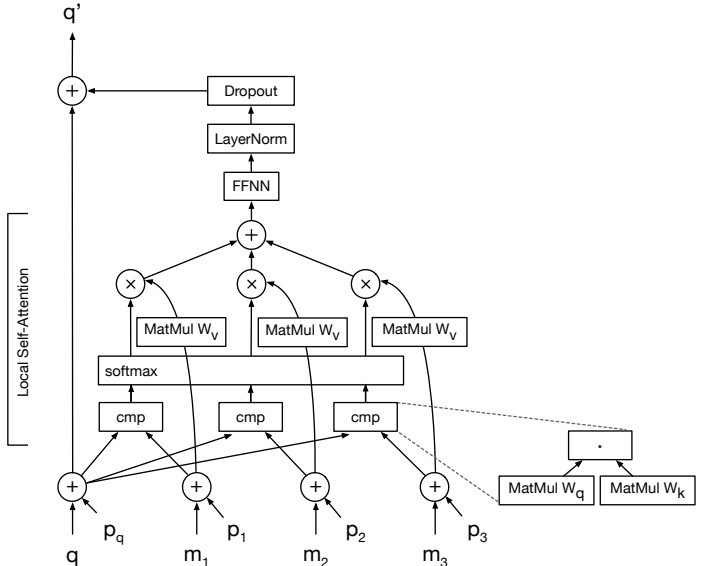

Figure 1: A slice of one layer of the Image Transformer, recomputing the representation $q'$ of a single channel of one pixel $q$ by attending to a memory of previously generated pixels $m_1, m_2, \ldots$. We apply a two-layer feed-forward neural network to the weighted average produced by the self-attention mechanism, perform layer normalization and sum the result with a residual connection. The position encodings $p_q, p_1, \ldots$ are added only in the first layer.

in opposition to a discriminator network trying to determine if a given image is real or generated. In contrast to the often blurry images generated by networks trained with likelihood-based losses, such generative adversarial networks (GANs) have been shown to generate sharper images with realistic high-frequency detail in generation and image super-resolution tasks (Zhang et al., 2016; Ledig et al., 2016).

While very promising, GANs have various drawbacks. They are notoriously unstable (Radford et al., 2015), motivating a large number of methods attempting to make their training more robust (Metz et al., 2016; Berthelot et al., 2017). Another common issue is that of mode collapse, where generated images fail to reflect the diversity in the training set (Metz et al., 2016).

A related problem is that GANs do not readily offer a probabilistic interpretation of their outputs, making it very challenging to measure the degree to which the models capture diversity. In contrast to models with a tractable likelihood, it also complicates optimizing model design, as objectively comparing different parameterizations or hyperparameter choices in this setting is considerably more difficult than comparing log-probabilities assigned to a validation set.

## 3 MODEL ARCHITECTURE

### 3.1 IMAGE REPRESENTATION AND 2D POSITIONAL INFORMATION

We treat both the input and predicted pixel RGB intensities as categorical variables rather than real numbers. Each input pixel's channel is encoded using a channel-specific set of $256$ $d$-dimensional embedding vectors of the channel intensity values $0 - 255$. For output intensities, we share a single, separate set of $256$ $d$-dimensional embeddings across channels.

We then combine the width and channels dimensions, yielding, for an image of width $w$ and height $h$, a 3-dimensional tensor with shape $[h, w \cdot 3, d]$.

To each pixel representation, we add a $d$-dimensional encoding of the coordinates of that pixel. Following Vaswani et al. (2017), the encoding consists of sine and cosine functions of the coordinates, with different frequencies across different dimensions. Since we need to represent two coordinates,

we use $d/2$ of the dimensions to encode the row number and the other $d/2$ of the dimensions to encode the the column and color channel.

The resulting tensor forms the input to our 2D local attention models (Section 3.3). For 1D local attention (Section 3.3) and the input to our super-resolution models we flatten this tensor in raster-scan order, similar to previous work (van den Oord et al., 2016). This yields a $[h \cdot w \cdot 3, d]$ tensor.

## 3.2 SELF-ATTENTION

Like the Transformer (Vaswani et al., 2017), the Image Transformer uses stacks of self-attention and position-wise feed-forward layers. Before we describe how we scale self-attention from sentences to images, which contain many more positions, we give a brief description of the self-attention layer.

Each self-attention layer computes a new $d$-dimensional representation for each position, that is each channel of each pixel. To recompute the representation for a given position, it first compares the position's current representation to other positions' representations, obtaining an attention distribution over the other positions. This distribution is then used to weight the contribution of the other postions' representations to the next representation for the position at hand.

Equation 1 and Figure1 fully describe all operations performed in every layer, independently for each position, with the exception of multi-head attention. For a detailed description of multi-head self-attention the reader is referred to (Vaswani et al., 2017).

$$q' = q + \text{dropout}(\text{layernorm}(\text{FFNN}(\text{softmax}\left(\frac{W_q q (MW_k)^T}{\sqrt{d}}\right) MW_v))) \tag{1}$$

In more detail, following previous work, we call the current representation of the pixel's channel, or position, to be recomputed the query $q$. The other positions whose representations will be used in computing a new representation for $q$ are $m_1, m_2, \ldots$ which together comprise the columns of the memory matrix $M$. Note that $M$ can also contain $q$. We first transform $q$ and $M$ linearly by learned matrices $W_q$ and $W_k$, respectively.

The self-attention mechanism then compares $q$ to each of the pixel's channel representations in the memory with a dot-product, scaled by $1/\sqrt{d}$. We apply the $\text{softmax}$ function to the resulting compatibility scores, treating the obtained vector as attention distribution over the pixel channels in the memory. After applying another linear transformation $W_v$ to the memory $M$, we compute a weighted average of the transformed memory, weighted by the attention distribution. In the decoders of our different models we mask the outputs of the comparisons appropriately so that the model cannot attend to positions in the memory that have not been generated, yet.

To the resulting vector we then apply a single-layer fully-connected feed-forward neural network with rectified linear activation followed by another linear transformation. The learned parameters of these are shared across all positions but different from layer to layer. Lastly, we perform layer normalization followed by dropout (Ba et al., 2016; Srivastava et al., 2014).

The entire self-attention operation can be implemented using highly optimized matrix multiplication code and executed in parallel for all pixels' channels.

## 3.3 LOCAL SELF-ATTENTION

The number of positions included in the memory $l_m$, or the number of columns of $M$, has tremendous impact on the scalability of the self-attention mechanism, which has a time complexity in $O(h \cdot w \cdot l_m \cdot d)$.

The encoders of our super-resolution models operate on $8 \times 8$ pixel images and it is computationally feasible to attend to all of their 192 positions. The decoders in our experiments, however, produce $32 \times 32$ pixel images with 3072 positions, rendering attending to all positions impractical.

Inspired by convolutional neural networks we address this by adopting a notion of locality, restricting the positions in the memory matrix $M$ to a local neighborhood around the query position. Changing this neighborhood per query position, however, would prohibit packing most of the computation

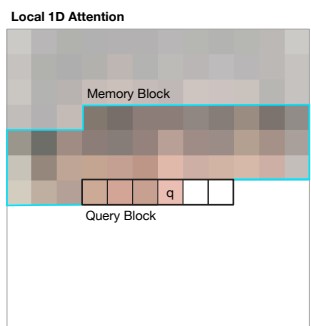 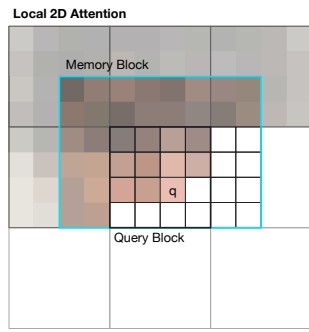

Figure 2: The two different conditional factorizations used in our experiments, with 1D and 2D local attention on the left and right, respectively. In both, the image is partitioned into non-overlapping query blocks, each associated with a memory block covering a superset of the query block pixels. In every self-attention layer, each position in a query block attends to all positions in the memory block. The pixel marked as $q$ is the last that was generated. All channels of pixels in the memory and query blocks shown in white have masked attention weights and do not contribute to the next representations of positions in the query block. While the effective receptive field size in this figure is the same for both schemes, in 2D attention the memory block contains a more evenly balanced number of pixels next to and above the query block, respectively.

necessary for self-attention into two matrix multiplications - one for computing the pairwise comparisons and another for generating the weighted averages. To avoid this, we partition the image into query blocks and associate each of these with a larger memory block that also contains the query block. For all queries from a given query block, the model attends to the same memory matrix, comprised of all positions from the memory block.

The self-attention is then computed for all query blocks in parallel, while the feed-forward networks and layer normalizations are computed in parallel for all positions.

In our experiments we use two different schemes for choosing query blocks and their associated memory block neighborhoods, resulting in two different factorizations of the joint pixel distribution into conditional distributions. Both are illustrated in Figure 2.

**1D Local Attention**   To compute self-attention on raster-scanned linearized images, we partition the length into non-overlapping query blocks $Q$ of length $l_q$, padding with zeroes if necessary. While contiguous in the linearized image, these blocks can be discontiguous in image coordinate space. For each query block we build the memory block $M$ from the same positions as $Q$ and an additional $l_m$ positions from pixels that have been generated before, which can result in overlapping memory blocks.

**2D Local Attention**   In 2D local attention models, we partition the image into query blocks rectangular and contiguous in the original image space. We generate the image one query block after another, ordering the blocks in raster-scan order. Within each block, we generate individual positions, or pixel channels, again in raster-scan order.

As illustrated in the right half of Figure 2, we generate the blocks outlined in grey lines left-to-right and top-to-bottom. We use 2-dimensional query blocks of a size $l_q$ specified by height and width $l_q = w_q \cdot h_q$, and memory blocks extending the query block to the top, left and right by $h_m, w_m$ and again $w_m$ pixels, respectively.

In both 1D and 2D local attention, we mask attention weights in the query and memory blocks such that positions that have not yet been generated are ignored.

As can be seen in Figure 2, 2D local attention balances horizontal and vertical conditioning context much more evenly. We believe this might have an increasingly positive effect on quality with growing image size as the conditioning information in 1D local attention becomes increasingly dominated by pixels next to a given position as opposed to above it.

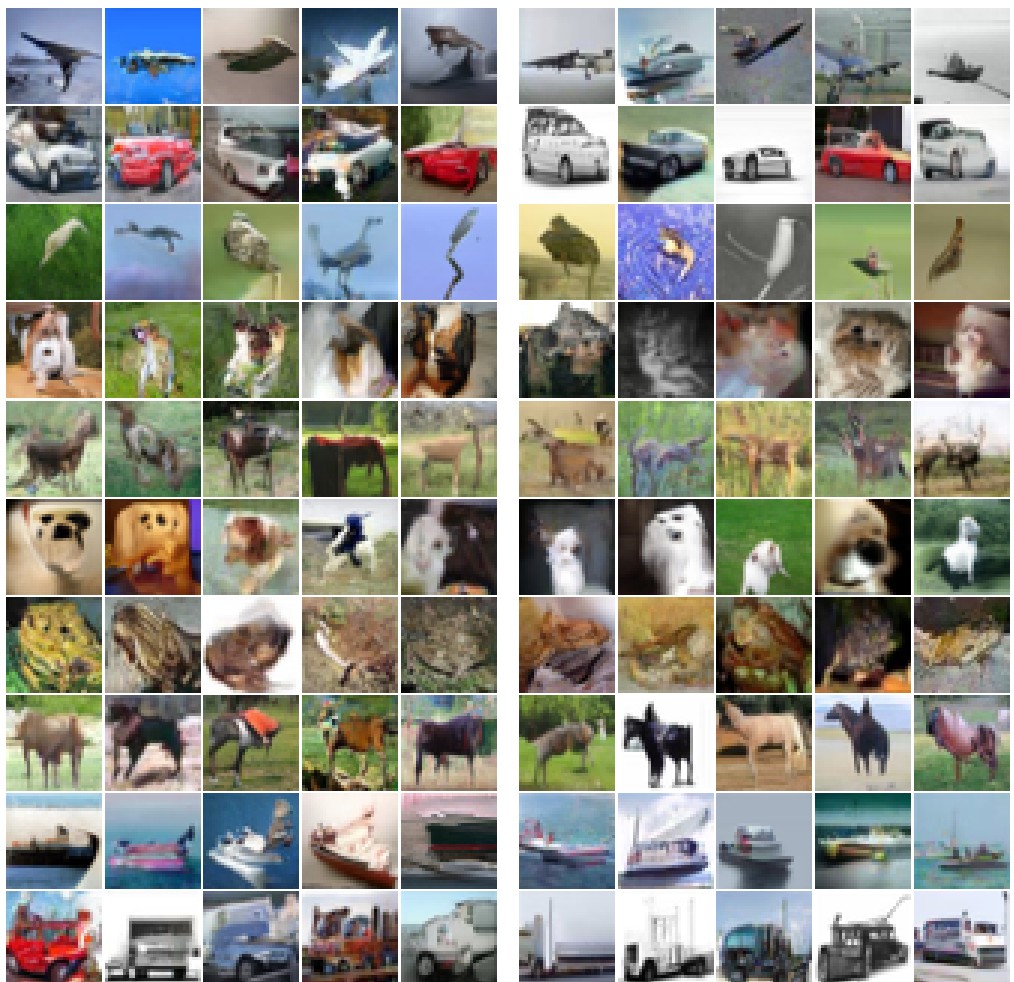

Table 2: Conditional image generations for all CIFAR-10 categories. Images on the left are from a model that achieves 3.03 bits/dim on the test set. Images on the right are from our best non-averaged model with 2.99 bits/dim. Both models are able to generate convincing cars, trucks, and ships. Generated horses, planes, and birds also look reasonable.

# 4 INFERENCE

Across all of the presented experiments, we sample from the various models with a tempered $\mathrm{softmax}$ (Dahl et al., 2017). We adjust the concentration of the distribution we sample from with a temperature $\tau > 0$ by which we divide the logits for the channel intensities.

We tuned $\tau$ between $0.8$ and $1.0$, observing the highest perceptual quality in unconditioned and class-conditional image generation with $\tau = 1.0$. For super-resolution we present results for different temperatures in Table 4.

# 5 EXPERIMENTS

For all of our experiments we optimize with Adam (Kingma & Ba, 2015), and vary the learning rate as specified in Vaswani et al. (2017). We train our models on both p100 and k40 GPUs, with batch sizes ranging from $1$ to $4$ per GPU.

Table 3: Negative log-likelihoods on the CIFAR-10 test and ImageNet validation sets. The Image Transformer outperforms all models but PixelCNN++, achieving a new state of the art on ImageNet. Larger memory blocks significantly improve its performance.

| Model Type | Memory Block Size | NLL | |
| --- | --- | --- | --- |
| | | CIFAR-10 (Test) | ImageNet (Validation) |
| Pixel CNN | - | 3.14 | - |
| Row Pixel RNN | - | 3.00 | 3.86 |
| Gated Pixel CNN | - | 3.03 | 3.83 |
| Pixel CNN++ | - | **2.92** | - |
| Image Transformer 1D local | 8 | 4.06 | - |
| | 16 | 3.47 | - |
| | 64 | 3.13 | - |
| | 256 | 2.99 | 3.78 |
| with checkpoint averaging | 256 | 2.98 | **3.77** |

## 5.1 GENERATIVE IMAGE MODELING

Our unconditioned and class-conditioned image generation models both use 1D local attention, with $l_q = 256$ and a total memory size of 512. On CIFAR-10 our best class-conditioned model (2.99 bits/dim) uses 8 self-attention and feed-forward layers, $d = 1024$, 16 attention heads, 2048 dimensions in the feed-forward layers, and a dropout of 0.3. Our smaller CIFAR-10 models (3.03 bits/dim) have $d = 512$, 1024 dimensions in the feed-forward layers, 8 attention heads and use $\mathrm{dropout} = 0.1$. Our state of the art ImageNet unconditioned generation model is significantly larger, with 12 self-attention and feed-forward layers, $d = 1024$, 4096-dimensional feed-forward layers, 16 attention heads, and $\mathrm{dropout} = 0.1$.

As Table 3 shows, our models improve over various previously proposed models including the PixelRNN and the gated PixelCNN. On ImageNet we establish a new state of the art of 3.78, which we can improve to 3.77 by averaging the last ten checkpoints.

While the PixelCNN++ achieved significantly better log-likelihoods on CIFAR-10 (Salimans et al.), we expect that many of the modifications in the PixelCNN++ carry over to the Image Transformer. We further believe our curated images for various classes to be of reasonable perceptual quality.

## 5.2 CONDITIONING ON IMAGE CLASS

We represent the image classes as learned $d$-dimensional embeddings per class and simply add the respective embedding to the input representation of every input position together with the positional encodings.

We trained the class-conditioned Image Transformer on CIFAR-10 and ImageNet data sets, achieving very similar log-likelihoods as in unconditioned generation. The perceptual quality of generated images, however, is significantly higher than that of our unconditioned models. We present some samples in Table 2.

## 5.3 IMAGE SUPER-RESOLUTION

Super-resolution is the process of recovering a high resolution image from a low resolution image while generating realistic and plausible details. Following (Dahl et al., 2017), in our experimental setup we enlarge an $8 \times 8$ pixel image four-fold to $32 \times 32$, a process that is massively underspecified: the model has to generate aspects such as texture of hair, makeup, skin and sometimes even gender that cannot possibly be recovered from the source image.

Here, we use the Image Transformer in an encoder-decoder configuration, connecting the encoder and decoder through an attention mechanism (Vaswani et al., 2017). Since the input is an $8 \times 8$ image, it is practical to use 1D attention with only one query and one memory block, each covering the entire image. We further use model dimension $d = 512$, 1024 hidden units in the position-wise

Table 4: Negative log-likelihood and human eval performance for the Image Transformer on CelebA. The fraction of humans fooled is significantly better than the previous state of the art. 2D local attention outperforms 1D local attention in the human evaluation.

| Model Type | NLL | %Fooled | | | |
|---|---|---|---|---|---|
| | $\tau = n/a$ | $\tau = 1.0$ | $\tau = 0.9$ | $\tau = 0.8$ |
| ResNet | - | 4.0 | | | |
| srez GAN (Garcia, 2016) | - | 8.5 | | | |
| PixelRecursive (Dahl et al., 2017) | - | - | 11.0 | 10.4 | 10.2 |
| ImageTransformer 1D local attention | 2.74 | | $21.5 \pm 4.0$ | $30.1 \pm 3.5$ | $32.5 \pm 3.0$ |
| ImageTransformer 2D local attention | 2.79 | | $31.25 \pm 3.5$ | $\mathbf{36.9} \pm 2.5$ | $32.5 \pm 2.5$ |

feed-forward network, 4 encoder layers and 12 decoder layers. We train end-to-end, maximizing likelihood.

For both of the following data sets, we resized the image to $8 \times 8$ pixels for the input and $32 \times 32$ pixels for the label using TensorFlow's `area` interpolation method.

**CelebA**   We trained on the standard CelebA data set of celebrity faces with cropped boundaries. Existing automated metrics like pSNR, SSIM and MS-SSIM have been shown to not correlate with perceptual image quality (Dahl et al., 2017). Instead, we conducted a human evaluation study on Amazon Mechanical Turk. Each worker is required to make a binary choice when shown one generated and one real image. Following (Dahl et al., 2017), we show 50 pairs of images, selected randomly, to 50 workers each. In our method, workers choose images from our model up to 36.9% of the time, a significant improvement over previous models. Sampling temperature of 0.9 and 2D local attention maximized perceptual quality as measured by this evaluation.

**CIFAR-10**   We also trained a super-resolution model on the CIFAR-10 data set. Our model reached a negative log-likelihood of 2.76 using 1D local attention and 2.78 using 2D local attention on the test set. As seen in Figure 5.3, our model commonly generates plausible looking objects even though the input images seem to barely show any discernible structure beyond coarse shapes.

## 6   CONCLUSION

In this work we demonstrate that models based on self-attention can operate effectively on modalities other than text, and through local self-attention scale to significantly larger structures than sentences. With fewer layers, its larger receptive fields allow the Image Transformer to improve over the state of the art in unconditional, probabilistic image modeling of comparatively complex images from ImageNet as well as super-resolution.

We further hope to have provided additional evidence that even in the light of generative adversarial networks, autoregressive generation of images is very much a promising area for further research - as is using network architectures such as the Image Transformer in GANs.

In future work we would like to explore a broader variety of conditioning information including free-form text, as previously proposed (Mansimov et al., 2015), and tasks combining modalities such as language-driven editing of images.

Fundamentally, we aim to move beyond still images to video (Kalchbrenner et al., 2016) and towards applications of such models in more model-based reinforcement learning approaches.

## REFERENCES

Jimmy Lei Ba, Jamie Ryan Kiros, and Geoffrey E Hinton. Layer normalization. *arXiv preprint arXiv:1607.06450*, 2016.

| Input | 1D Local Attention | | | 2D Local Attention | | | Original |
|---|---|---|---|---|---|---|---|
| | $\tau = 0.8$ | $\tau = 0.9$ | $\tau = 1.0$ | $\tau = 0.8$ | $\tau = 0.9$ | $\tau = 1.0$ | |

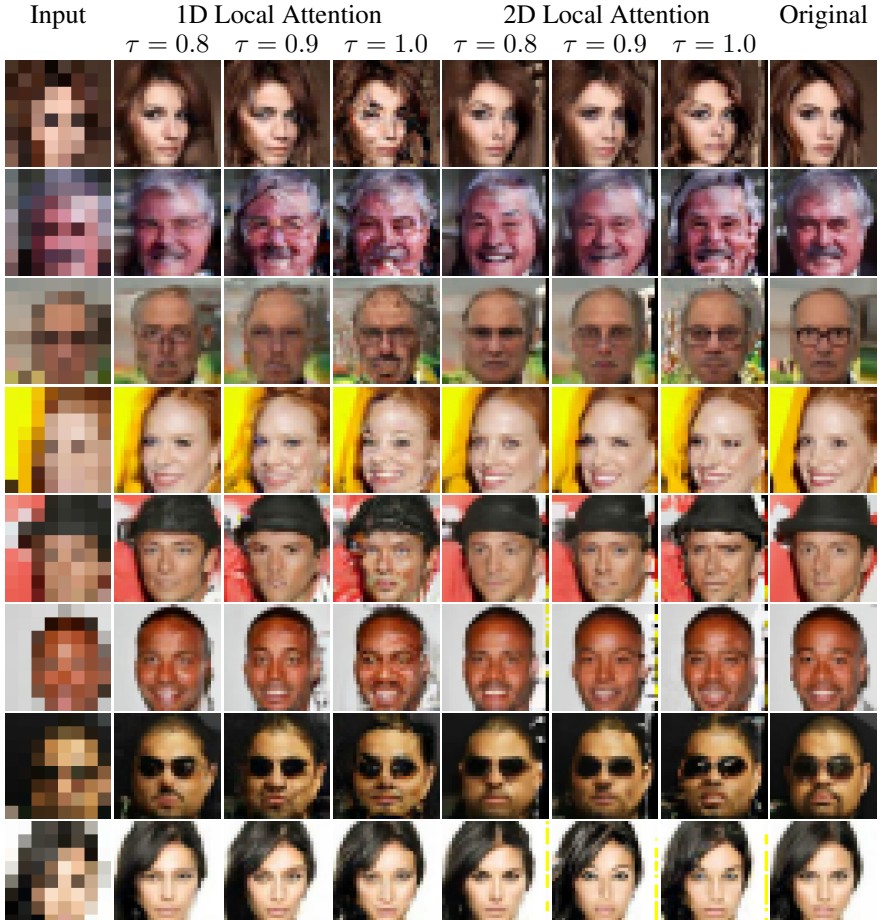

Table 5: Images from our 1D and 2D local attention super-resolution models trained on CelebA, sampled with different temperatures. 2D local attention with $\tau = 0.9$ scored highest in our human evaluation study.

Marc G. Bellemare, Sriram Srinivasan, Georg Ostrovski, Tom Schaul, David Saxton, and Rémi Munos. Unifying count-based exploration and intrinsic motivation. *CoRR*, abs/1606.01868, 2016. URL http://arxiv.org/abs/1606.01868.

Yoshua Bengio and Samy Bengio. Modeling high-dimensional discrete data with multi-layer neural networks. In *ADVANCES IN NEURAL INFORMATION PROCESSING SYSTEMS 12*, pp. 400–406. MIT Press, 2000.

David Berthelot, Tom Schumm, and Luke Metz. BEGAN: boundary equilibrium generative adversarial networks. *CoRR*, abs/1703.10717, 2017. URL http://arxiv.org/abs/1703.10717.

Jianpeng Cheng, Li Dong, and Mirella Lapata. Long short-term memory-networks for machine reading. *arXiv preprint arXiv:1601.06733*, 2016.

Ryan Dahl, Mohammad Norouzi, and Jonathan Shlens. Pixel recursive super resolution. 2017. URL https://arxiv.org/abs/1702.00783.

David Garcia. srez: Adversarial super resolution. https://github.com/david-gpu/srez, 2016. URL https://github.com/david-gpu/srez.

Ian Goodfellow, Jean Pouget-Abadie, Mehdi Mirza, Bing Xu, David Warde-Farley, Sherjil Ozair, Aaron Courville, and Yoshua Bengio. Generative adversarial nets, 2014.

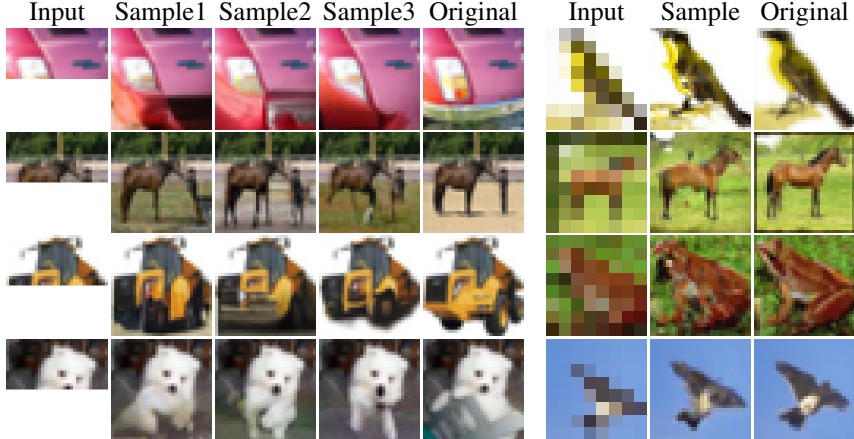

| Input | Sample1 | Sample2 | Sample3 | Original | Input | Sample | Original |

Table 6: On the left are image completions from our best conditional generation model, where we sample the second half. On the right are samples from our four-fold super-resolution model trained on CIFAR-10. Our images look realistic and plausible, show good diversity among the completion samples and observe the outputs carry surprising details for coarse inputs in super-resolution.

Nal Kalchbrenner and Phil Blunsom. Recurrent continuous translation models. In *Proceedings EMNLP 2013*, pp. 1700–1709, 2013. URL http://nal.co/papers/KalchbrennerBlunsom_EMNLP13.

Nal Kalchbrenner, Aäron van den Oord, Karen Simonyan, Ivo Danihelka, Oriol Vinyals, Alex Graves, and Koray Kavukcuoglu. Video pixel networks. *CoRR*, abs/1610.00527, 2016. URL http://arxiv.org/abs/1610.00527.

Diederik Kingma and Jimmy Ba. Adam: A method for stochastic optimization. In *ICLR*, 2015.

Hugo Larochelle and Iain Murray. The neural autoregressive distribution estimator. In *The Proceedings of the 14th International Conference on Artificial Intelligence and Statistics*, volume 15 of *JMLR: W&CP*, pp. 29–37, 2011.

Christian Ledig, Lucas Theis, Ferenc Huszar, Jose Caballero, Andrew Aitken, Alykhan Tejani, Johannes Totz, Zehan Wang, and Wenzhe Shi. Photo-realistic single image super-resolution using a generative adversarial network. *arXiv:1609.04802*, 2016.

Elman Mansimov, Emilio Parisotto, Lei Jimmy Ba, and Ruslan Salakhutdinov. Generating images from captions with attention. *CoRR*, abs/1511.02793, 2015. URL http://arxiv.org/abs/1511.02793.

Luke Metz, Ben Poole, David Pfau, and Jascha Sohl-Dickstein. Unrolled generative adversarial networks. *CoRR*, abs/1611.02163, 2016. URL http://arxiv.org/abs/1611.02163.

Ankur Parikh, Oscar Tckstrm, Dipanjan Das, and Jakob Uszkoreit. A decomposable attention model. In *Empirical Methods in Natural Language Processing*, 2016. URL https://arxiv.org/pdf/1606.01933.pdf.

Alec Radford, Luke Metz, and Soumith Chintala. Unsupervised representation learning with deep convolutional generative adversarial networks. *CoRR*, abs/1511.06434, 2015. URL http://arxiv.org/abs/1511.06434.

Tim Salimans, Andrej Karpathy, Xi Chen, Diederik P. Kingma, and Yaroslav Bulatov. Pixelcnn++: A pixelcnn implementation with discretized logistic mixture likelihood and other modifications. under review at ICLR 2017.

Nitish Srivastava, Geoffrey E Hinton, Alex Krizhevsky, Ilya Sutskever, and Ruslan Salakhutdinov. Dropout: a simple way to prevent neural networks from overfitting. *Journal of Machine Learning Research*, 15(1):1929–1958, 2014.

Lucas Theis and Matthias Bethge. Generative image modeling using spatial lstms. In *Proceedings of the 28th International Conference on Neural Information Processing Systems - Volume 2*, NIPS'15, pp. 1927–1935, Cambridge, MA, USA, 2015. MIT Press. URL `http://dl.acm.org/citation.cfm?id=2969442.2969455`.

Aäron van den Oord and Benjamin Schrauwen. The student-t mixture as a natural image patch prior with application to image compression. *Journal of Machine Learning Research*, 15:2061–2086, 2014. URL `http://jmlr.org/papers/v15/vandenoord14a.html`.

Aäron van den Oord, Nal Kalchbrenner, and Koray Kavukcuoglu. Pixel recurrent neural networks. *ICML*, 2016.

Ashish Vaswani, Noam Shazeer, Niki Parmar, Jakob Uszkoreit, Llion Jones, Aidan N. Gomez, Lukasz Kaiser, and Illia Polosukhin. Attention is all you need. 2017. URL `http://arxiv.org/abs/1706.03762`.

Han Zhang, Tao Xu, Hongsheng Li, Shaoting Zhang, Xiaolei Huang, Xiaogang Wang, and Dimitris N. Metaxas. Stackgan: Text to photo-realistic image synthesis with stacked generative adversarial networks. *CoRR*, abs/1612.03242, 2016. URL `http://arxiv.org/abs/1612.03242`.

## A  CELEBA SUPERRESOLUTION

Image pairs comparing ratings of generated images by the Local 2D ImageTransformer model and the original images. On the left side are images where the raters prefer the generated image over the original ones. On the right side, raters prefer the original over generated image.

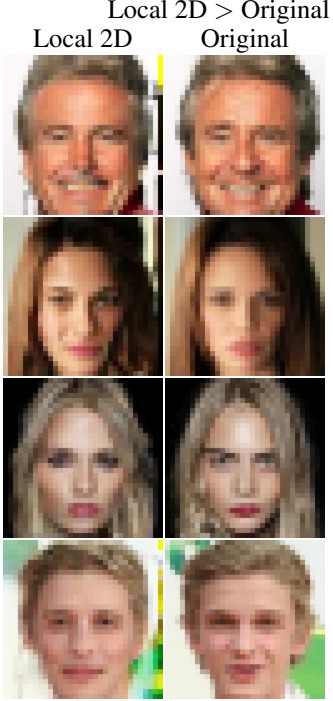
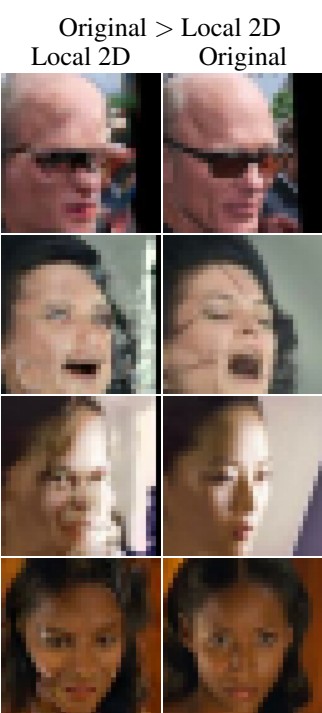

