# OpenReview forum: "Image Transformer"
_ICLR.cc/2018/Conference — Reject_

### Official Review · AnonReviewer1 · 2017-11-26
**Interesting idea, weak evaluation**

**Rating:** 6
**Confidence:** 4

**Review:**

Summary

This paper extends self-attention layers (Vaswani et al., 2017) from sequences to images and proposes to use the layers as part of PixelCNNs (van den Oord et al., 2016). The proposed model is evaluated in terms of visual appearance of samples and log-likelihoods. The authors find a small improvement in terms of log-likelihood over PixelCNNs and that super-resolved CelebA images are able to fool human observers significantly more often than PixelRNN based super-resolution (Dahl et al., 2017).

Review

Autoregressive models are of large interest to the ICLR community and exploring new architectures is a valuable contribution. Using self-attention in autoregressive models is an intriguing idea. It is a little bit disappointing that the added model complexity only yields a small improvement compared to the more straight-forward modifications of the PixelCNN++. I think the paper would benefit from a little bit more work, but I am open to adjusting my score based on feedback.

I find it somewhat surprising that the proposed model is only slightly better in terms of log-likelihood than a PixelRNN, but much better in terms of human evaluation – given that both models were optimized for log-likelihood. Was the setup used with Mechanical Turk exactly the same as the one used by Dahl et al.? These types of human evaluations can be extremely sensitive to changes in the setup, even the phrasing of the task can influence results. E.g., presenting images scaled differently can mask certain artifacts. In addition, the variance between subjects can be very high. Ideally, each method included in the comparison would be re-evaluated using the same set of observers. Please include error bars.

The CelebA super-resolution task furthermore seems fairly limited. Given the extreme downsampling of the input, the task becomes similar to simply generating any realistic image. A useful baseline would be the following method: Store the entire training set. For a given query image, look for the nearest neighbor in the downsampled space, then return the corresponding high-resolution image. This trivial method might not only perform well, it also highlights a flaw in the evaluation: Any method which returns stored high-resolution images – even if they don’t match the input – would perform at 50%. To fix this, the human observers should also receive the low-resolution image and be asked to identify the correct corresponding high-resolution image.

Using multiplicative operations to model images seems important. How does the self-attention mechanism relate to “gated” convolutions used in PixelCNNs? Could gated convolutions not also be considered a form of self-attention?

The presentation/text could use some work. Much of the text assumes that the reader is familiar with Vaswani et al. (2017) but could easily be made more self-contained by directly including the definitions used. E.g., the encoding of positions using sines and cosines or the multi-head attention model. I also felt too much of the architecture is described in prose and could be more efficiently and precisely conveyed in equations.

On page 7 the authors write “we believe our cherry-picked images for various classes to be of higher perceptual quality”. This is a meaningless result, not only because the images were cherry-picked. Generating realistic images is trivial - you just need to store the training images. Analyzing samples generated by a generative model (outside the context of an application) should therefore only be used for diagnostic purposes or to build intuitions but not to judge the quality of a model.

Please consider rephrasing the last sentence of the abstract. Generating images which “look pretty cool” should not be the goal of a serious machine learning paper or a respected machine learning conference.

---

> ### Author Response · Authors · 2017-12-23
> **Thank you for your review. Please find our responses below.**
>
> We thank the reviewer for the thorough and insightful review.
> Reviewer:
> ...surprising that the proposed model is only slightly better in terms of log-likelihood ... Was the setup used ... exactly the same as the one used by Dahl et al.?
>
> Our response:
> Our generation models had not finished training on ImageNet. We now have significantly better perplexities (3.78 bits/dim, 3.77 with checkpoint averaging) than the row PixelRNN (3.86 bits/dim) and the Gated PixelCNN (3.83 bits/dim, previous SOTA) models. Gated PixelCNN improved over the previous SOTA by only 0.03, our improvement is twice as large.
>
> Reviewer:
> … human evaluations can be extremely sensitive to changes in the setup … Please include error bars.
>
>
> Our response:
> We included error bars which show that the variance is small, and the subjects (50 per image) are fairly clear on their preferences of images. We ensured that we use the exact same evaluation setup, down to the interface presented to the subjects.
>
> Reviewer:
> The CelebA super-resolution task … to fix this, the human observers should also receive the low-resolution image
>
> Our response:
> We agree, yet followed Dahl, et al.’s procedure exactly for comparability. While the shortcoming of the evaluation does present a potential loophole, allowing the model to generate images that do not downsample back to the input image, our model does not exploit this. It does generate images that, when downsampled, yield images very close to the low resolution input.
>
> To demonstrate this, we compared the pixel/channel-level L2^2-distance between the low-resolution input image and the downsampled output image. Across 300 images from the CelebA test set, the average per-pixel, per-channel distance in normalized intensities between the input and the downsampled output images is 0.0106. The average distance between each of the low-resolution input images and 100 other downsampled images from the CelebA test set each is 0.1188. Given these are all cropped images of faces, we believe the difference to be significant. To underline this, we chose those two input images for which the downsampled version of the output image generated by our model is most different from the input image according to this metric and made them available here. Even here the downsampled output of our model is very similar to the original input.
>
> Due to shortage of space, we kindly request the reviewer to refer to the links in our response to Anonymous Reviewer 2.
>
> The respective distances are 0.0344 and 0.03119 (between the input and model output images, each downsampled). We hope this shows that our model generates plausible images adhering to the intended constraint: that when downsampled they are very similar to the original low-resolution input image.
>
> The input images constitute rich conditioning information such as hair and skin color, object position and pose, background color, etc. We believe there is real demand for models improving perceptual detail in images without a specific, expected output.
>
>
> Reviewer:
> ...Could gated convolutions not also be considered a form of self-attention?
>
>
>
> Our response:
> There is some similarity to the multiplicative effects in self attention and gated CNNs, but there are also clear differences. Both use gating to scale the activations in multiplicative terms, which can prevent gradients from ‘blowing up’.
>
> In self-attention, we have two sources of multiplicative interactions: 1) softmax-gated query key inner products which give us multiplicative effects between query and key representations, and 2) multiplicative interaction of the softmax weights with all values at each position, once per head. In self-attention, we first filter (softmax gating), and then aggregate (linear combination of values) per head, while gated convolutions first aggregate (applying the kernel) and then filter (gating). Because of the large receptive field of local self-attention, we achieve multiplicative interactions between positions that are far apart, which can be computationally expensive for convolutions. Both gating mechanisms are complementary and can be used together, e.g. the gating from gated PixelCNN could replace our position-wise FFNN layers.
>
> Reviewer:
> The presentation/text could use some work. ….
>
> Our response:
> We hoped to fit the submission within 9 pages, focusing on novel content at the expense of being self-contained. However, Equation 1 describes the computation applied to each position in every layer completely - the only exception being multiple heads. If accepted, we will repeat more unchanged details of the model.
>
> Reviewer:
> The authors write “we believe our cherry-picked images for various classes to be of higher perceptual quality”.
>
> Our response:
> We have revised our language and now write “we believe our curated images for various classes to be of reasonable perceptual quality”, without comparing the perceptual quality to other work.
>
> We removed the statement on “pretty cool” images from the abstract.

---

### Official Review · AnonReviewer3 · 2017-11-29
**Requires improvement**

**Rating:** 3
**Confidence:** 3

**Review:**

This paper extends the PixelCNN/RNN based (conditional) image generation approaches with self-attention mechanism.

Pros:
- qualitatively the proposed method has good results in several tasks

Cons:
- writing needs to be improved
- lack of motivation
- not easy to follow technique details


The motivation part is missing. It seems to me that the paper simply try to combine the Transformer with PixelCNN/RNN based image generation without a clear explanation why this is needed. Why self-attention is so important for image generation? Why not just a deeper network with more parameters? Throughout the paper I cannot find a clear answer for this. Based on this I couldn't see a clear contribution.

The paper is difficult to keep the track given the current flow. Each subsection of section 3 starts with technique details without explaining why we do this. Some sentences like "look pretty cool" is not academic.

The experiments lack comparisons except the human evaluation, while the log-likelihood improvement is marginal. I am wondering how the human evaluation is conducted. Does it compare all the competing algorithms against the same sub-samples of the GT data? How many pairs have been compared for each algorithm?  Apart from this metric, I would like to see qualitative comparison between competing algorithms in the paper as well. Also other approaches e.g. SRGAN could be compared.

I am also interested about the author's claim that the implementation error that influences the log-likelihood. Has this been fixed after the deadline?

---

> ### Author Response · Authors · 2017-12-23
> **Thank you for your review. Please find our responses below**
>
> We thank the reviewer for your insightful review.
>
> At the time of submission, our conditional and unconditional generation generation models had not finished training on ImageNet, the harder task. We now have even better perplexities (3.78 bits/dim, 3.77 with checkpoint averaging) than the row PixelRNN (3.86 bits/dim) and the Gated PixelCNN (3.83 bits/dim, previous state of the art) models.
> Gated PixelCNN improved over the previous state of the art by only 0.03, while our improvement is twice as large. Over the entire image of 3072 dimensions, the improvement in bits is quite significant.
>
> Reviewer:
> The motivation part is missing. ... Why not just a deeper network with more parameters? Throughout the paper I cannot find a clear answer for this. Based on this I couldn't see a clear contribution.
>
> Our response:
> We agree with the reviewer that our motivation might not have been described well in the original submission. We added a more thorough motivation to the introduction of the paper, which we repeat here in summary for your convenience.
>
> One disadvantage of CNNs compared to RNNs is their typically fairly limited receptive field. This can adversely affect their ability to model long-range phenomena common in images, such as symmetry and occlusion, especially with a small number of layers. Growing the receptive field, like deepening the network, however, comes at a significant cost in number of parameters and hence computation and can make training such models more challenging.
>
> With the Image Transformer, we aim to find a better balance in the trade-off between the virtually unlimited receptive field of the necessarily sequential PixelRNN and the limited receptive field of the much more
> parallelizable PixelCNN and its various extensions.
>
> We propose eschewing recurrent and convolutional networks in favor of a model based entirely on a locally restricted form of multi-head self-attention that could also be interpreted as a sparsely parameterized form of convolution,  allowing  for  significantly  larger  receptive  fields  than  CNNs  at  the  same  number  of parameters.
>
> We furthermore added additional experiments indicating that indeed, increasing the size of the receptive field significantly improves the performance of our model, allowing it to (now significantly, see below) outperform the state of the art. These show that increasing the perceptive field from 16 to 256 positions, for instance, improves perplexity on CIFAR-10 Test from 3.47 to 2.99.
>
>
> Reviewer:
> The paper is difficult to keep the track given the current flow. Each subsection of section 3 starts with technique details without explaining why we do this. Some sentences like "look pretty cool" is not academic.
>
> Our response:
> We removed that sentence from the abstract, added additional material on the motivation, as summarized above, and tried to improve the overall flow in a flew places.
>
>
> Reviewer:
> The experiments lack comparisons except the human evaluation,  … Does it compare all the competing algorithms against the same sub-samples of the GT data? ...
>
> Our response:
> We follow the same evaluation procedure as Dahl, et al.’s paper but do not use the same exact sub-samples as we could not recover them. For each model, we use 50 randomly selected dev images where each image is rated by 50 workers. We use a different set of workers for each model. Also, our latest results on imagenet unconditional generation perplexities show a significant improvement in log likelihoods over the previous state-of-the-art.
>
> Reviewer:
> Apart from this metric, I would like to see qualitative comparison between competing algorithms in the paper as well. Also other approaches e.g. SRGAN could be compared.
>
> Our Response:
> It would be very difficult to conduct a proper qualitative evaluation because we are missing representative samples from the various algorithms. We hope that our human evaluation numbers capture some qualitative differences between our model and pixel cnn.
>
> Reviewer:
> I am also interested about the author's claim that the implementation error that influences the log-likelihood. Has this been fixed after the deadline?
>
> Our response:
> We have indeed fixed the bug. The resulting images are now free of artifacts and the log-likelihood did improve. That said, we are still in the middle of conducting a final, apples-to-apples comparison between 1D and 2D self attention on various super-resolution tasks and will include the results of this in the final paper.

---

### Official Review · AnonReviewer2 · 2017-12-06
**Missing motivation and mathematical details**

**Rating:** 5
**Confidence:** 4

**Review:**

In this paper the authors propose an autoregressive image generation model that incorporates a self-attention mechanism. The latter is inspired by the work of [Vaswani et al., 2016], which was proposed for sequences and is extended to 2D images in this work. The authors apply their model to super-resolution of face images, as well as image completion (aka inpainting) and generation, both unconditioned or conditioned on one of a small number of image classes from the CIFAR10 and ImageNet datasets. The authors evaluate their method in terms of visual quality of their generated images via an Amazon Mechanical Turk survey and quantitatively by reporting slightly improved log-likelihoods.

While the paper is well written, the motivation for combining self-attention and autoregressive models remains unclear unfortunately, even more though as the reported quantitative improvement in terms of log-likelihood are only marginal. The technical exposition is at times difficult to follow with some design decisions of the network layout being quite ad hoc and not well motivated. Expressing the involved operations in mathematical terms would help comprehend some of the technical details and add to the reproducibility of the proposed model.

Another concern is the experimental evaluation. While the reported log-likelihoods are only marginally better, the authors report a significant boost in how often humans are fooled by the generated images. While the image generation is conditioned on the low-resolution input, the workers in the Amazon Mechanical Turk study get to see the high-resolution images only. Of course, a human observer would pick the one image out of the two shown images which is more realistic although it might have nothing to do with the input image, which seems wrong. Instead, the workers should see the low-res input image and then have to decide which high-res image seems a better match or more likely.

Overall, the presented work looks quite promising and an interesting line of research. However, in its present form the manuscript doesn't seem quite ready for publication yet. Though, I would strongly encourage the authors to make the exposition more self-contained and accessible, in particular through rigorous mathematical terms, which would help comprehend the involved operations and help understand the proposed mechanism.

Additional comments:
- Abstract: "we also believe to look pretty cool". Please re-consider the wording here. Generating "pretty cool" images  should not be the goal of a scientific work.

---

> ### Author Response · Authors · 2017-12-23
> **Thank you for your review. Please find our responses below.**
>
> We thank the reviewer for your review.
> At submission time, our generation models had not finished training on ImageNet, the harder task. We now have significantly better perplexities (3.78 bits/dim, 3.77 with checkpoint averaging) than the row PixelRNN (3.86 bits/dim) and the Gated PixelCNN (3.83 bits/dim, previous SOTA) models.
> Gated PixelCNN improved over the previous SOTA by only 0.03, while our improvement is twice as large. The improvement in bits over an entire image with 3072 positions is significant.
>
> Reviewer:
> While the paper is well written, …. some design decisions of the network layout being quite ad hoc and not well motivated.
>
> Response:
> We agree that our motivation was not sufficiently described in our original submission. We added a much more detailed motivation in the introduction, summarized here for convenience.
>
> A disadvantage of CNNs compared to RNNs is their typically limited receptive field. This can adversely affect their ability to model long-range phenomena common in images, such as symmetry and occlusion in a small number of layers. Growing the receptive field, or deepening the network, comes at great cost in number of parameters and computation and can make training such models harder.
>
> In this work we aim to find a better balance in the trade-off between the virtually unlimited receptive field of the necessarily sequential PixelRNN and the limited receptive field of the much more parallelizable PixelCNN.
>
> The locally restricted form of multi-head self-attention we propose could also be interpreted as a sparsely parameterized form of convolution, with significantly larger receptive fields than CNNs at the same number of parameters.
>
> We added experimental results that indicating that indeed, increasing the size of the receptive field improves the performance of our model significantly.
>
>
> Reviewer:
> Expressing the involved operations in mathematical terms would help comprehend ...
>
> Our response:
> We agree, though Equation 1 does describe the computation performed per position in the each of the layers completely, with the only exception being multiple heads. If the paper is accepted, we will elaborate more on the details of the model to make the content more self-contained, repeating the equations for multi-head attention and the positional encodings, etc.
>
> Reviewer:
> Another concern is the experimental evaluation....
>
> Our response:
> We agree, but decided to exactly follow Dahl, et al.’s procedure for comparability. While this shortcoming does present a potential loophole by allowing the model to generate images that do not downsample back to the input image (which we consider to be rich conditioning), our model does not exploit this, but instead generates images that, when downsampled again, yield images very close to the low resolution input.
>
> To demonstrate this, we conducted an analysis comparing the pixel/channel-level L2^2-distance between the low-resolution input image and the downsampled output image.
> Across a set of 300 images from the CelebA test set, the average per-pixel, per-channel distance in normalized intensities between the input and the downsampled output images is 0.0106. The average distance between the each of the low-resolution input images and 100 other downsampled images from the CelebA test set each is 0.1188. Given that these are all cropped images of faces, we believe the difference to be significant. To underline this, we chose those two input images for which the downsampled version of the output image generated by our model is most different from the input image according to this metric and made them available. The downsampled output of our model is still very similar to the original, downsampled input.
>
> Example 1
> original input image:  http://tiny.cc/kq8mpy
> downsampled input image:  http://tiny.cc/xq8mpy
> super-resolved generated image (generated by the model):  http://tiny.cc/7r8mpy
> downsampled super-resolved generated image:  http://tiny.cc/gs8mpy
>
> Example 2
> original input image:  http://tiny.cc/kt8mpy
> downsampled input image: http://tiny.cc/tt8mpy
> super-resolved generated image (generated by the model):  http://tiny.cc/5t8mpy
> downsampled super-resolved generated image:  http://tiny.cc/eu8mpy
>
> The respective distances are 0.0344 and 0.03119 (between the downsampled original and the downsampled model output). We hope this persuades the reviewer that our model generates plausible images that adhere to the intended constraint: that they, when downsampled, are very similar to the original low-resolution input image.
>
> Reviewer:
> ...strongly encourage the authors to make the exposition more self-contained and accessible, ….
>
> Our Response:
> We hope the additional motivation helped improve the accessibility. If the paper is accepted, we will be happy to elaborate more on the details of the model from Vaswani et al. (2017) to make the content more self-contained.
>
> We removed the statement about generating “pretty cool” images from the abstract.

---

### Decision · Program_Chairs · 2018-01-29
**ICLR 2018 Conference Acceptance Decision**

**Decision:**

Reject

**Comment:**

This paper had some quality and clarity issues and the lack of motivation for the approach was pointed out by multiple reviewers.  Just too far away from the acceptance threshold.